# Detection of Lymphatic Vessels in Dental Pulp

**DOI:** 10.3390/biology11050635

**Published:** 2022-04-21

**Authors:** Kamila Wiśniewska, Zbigniew Rybak, Maria Szymonowicz, Piotr Kuropka, Katarzyna Kaleta-Kuratewicz, Maciej Dobrzyński

**Affiliations:** 1Department of Dental Surgery, Wroclaw Medical University, Krakowska 26, 50-425 Wroclaw, Poland; 2Pre-Clinical Research Centre, Wroclaw Medical University, Bujwida 44, 50-345 Wroclaw, Poland; zbigniew.rybak@umw.edu.pl (Z.R.); maria.szymonowicz@umw.edu.pl (M.S.); 3Department of Histology and Embryology, Wroclaw University of Environmental and Life Sciences, Norwida 25, 50-375 Wroclaw, Poland; piotr.kuropka@upwr.edu.pl (P.K.); katarzyna.kaleta-kuratewicz@upwr.edu.pl (K.K.-K.); 4Department of Pediatric Dentistry and Preclinical Dentistry, Wroclaw Medical University, Krakowska 26, 50-425 Wroclaw, Poland; maciej.dobrzynski@umw.edu.pl

**Keywords:** histology, human dental pulp, immunohistochemistry, lymphatic and blood markers, lymphatic vessels

## Abstract

**Simple Summary:**

Our studies and results suggest a moderate correlation between pulp inflammation and the formation of new vessels, including lymphatic vessels. In view of the above, it can be concluded that as inflammation increases, the size of the vascular bed that enables circulation of body fluids, blood, and lymph increases as well. However, microscopic and immunohistochemical studies did not conclusively demonstrate if these vessels form systems within the pulp that facilitate drainage of the tooth cavity.

**Abstract:**

The literature lacks conclusive evidence that lymphatic vessels can form in the dental pulp. Lymphangiogenesis is believed to occur in an inflamed pulp. If one defines lymphangiogenesis as the development of lymphatic vessels from already existing ones, such a mechanism is possible only when lymphatic vessels are present in healthy teeth. This paper aims to identify lymphatic vessels in the dental pulp using microscopic and immunohistochemical methods under physiological and pathological conditions. The tissue material consisted of human teeth intended for extraction. Our studies and results suggest a moderate correlation between pulp inflammation and the formation of new vessels, including lymphatic vessels.

## 1. Introduction

There are two ways of lymphatic vessel development. The first is endothelial cell differentiation into lymphatic vascular endothelial cells. The second is the separation of newly formed lymphatic vessels from already existing ones, the so-called lymphangiogenesis [1,2,3]. Lymphangiogenesis observed in an inflamed dental pulp is the response to inflammatory stimuli in equine molars [4]. Unfortunately, the study did not reveal whether lymphatic vessels can be formed anew in the dental pulp. However, this phenomenon was observed in the cornea of adult mice [5].

The presence of lymphatic vessels in the dental pulp has been investigated in many species of animals and humans. Studies on lymphatic vessels involving light microscopes, which date back to the early 20th century, determined morphological features of lymphatic vessels, namely thin walls with irregular shapes and the absence of erythrocytes in the vessel lumen. However, these methods are non-specific. The immunohistochemical method is very likely to reveal the presence of a lymphatic system in dental tissues. This method uses labelled antibodies against antigens typical of lymphatic vessels [6].

Studies involving a transmission electron microscope confirmed the presence of lymphatic vessels in the dental pulp based on a characteristic structure of light and the profile of lymphatic vessels in inflamed dental pulp [7,8]. Lymphatic vessels in the tooth were detected using enzymatic and histochemical staining with 5′-nucleotidase and immunohistochemical staining [9,10]. The literature lacks conclusive evidence that lymphatic vessels can form in the dental pulp. Martin et al. refuted the presence of lymphatic vessels in the dental pulp and suggested that capillaries and interstitial tissue could create a closed circulatory system without lymphatic vessels [3]. Gerli et al. [11] observed that detection of lymphatic vessels is likely false positive resulting from fibre bundles and inflammatory cells stained due to methodological errors. Blood circulation within the alveolus is of great importance in maintaining the homeostasis of the dental pulp and the nerves and odontoblasts contained in it. The lymphatic system constitutes an alternative for recirculating intercellular fluids. It is also an element of defence against potential bacterial infection associated with pathological processes in the oral cavity. Understanding angiogenesis and lymphangiogenesis during these processes can contribute to more effective disease treatment, although further studies are needed. Lymphangiogenesis is believed to occur in an inflamed dental pulp. If one defines lymphangiogenesis as the development of lymphatic vessels from already existing ones, such a mechanism is possible only when lymphatic vessels are present in healthy teeth. This paper aims to demonstrate lymphatic vessels in the dental pulp under physiological conditions and in teeth with caries-induced inflammation using microscopic and immunohistochemical methods.

## 2. Materials

The tissue material were human teeth obtained from the dental clinic. Indications for tooth extraction included:-Extensive destruction of the crown and caries;-Orthodontic reasons, caries-free teeth.

The number of teeth used in the study is given in Table 1.

The teeth were divided into four groups:Group I—healthy teeth.Group II—teeth with neutrophilic infiltration.Group III—teeth with mixed lymphocytic and neutrophilic infiltration.Group IV—teeth with wide and numerous blood-filled vessels within the dental pulp and without a significant number of inflammatory cells.

Group I teeth were obtained from patients in whom tooth extraction was not related to inflammation; the teeth were extracted for orthodontic reasons. Groups II and III comprised untreated teeth with caries removed due to significant tissue damage. Group IV primarily included teeth previously treated for caries that were later extracted due to extensive tissue loss.

This study was conducted at the Department of Experimental Surgery and Biomaterials Research, Wroclaw Medical University, and the Department of Animal Anatomy and Physiology, Wroclaw University of Life Sciences. It was approved by the Bioethics Committee of the Wroclaw Medical University (KB-796/2017).

## 3. Methods

### 3.1. Light and Fluorescence Microscope Examination

The materials for histological examination, i.e., human teeth, were fixed in 4% buffered formalin (pH = 7.2) and then decalcified in 10% ethylenediaminetetraacetic acid (EDTA) solution and a mixture of 5% sodium citrate with 5% formic acid. After reaching the tissue softness threshold, the teeth were rinsed for 24 h in running water, and the samples were dehydrated in an alcohol series and embedded in paraffin. Paraffin blocks were cut on a microtome into 4 µm slices, which were deparaffinised. After rehydration, haematoxylin-eosin (HE) staining was performed. The prepared samples were evaluated using an Eclipse 80imicroscope (Nikon, Tokyo, Japan) under transmitted light. For fluorescence examinations, 10% acridine orange (Sigma-Aldrich, Darmstadt, Germany) was used to stain the strips. The material was analysed using an Eclipse 80i research microscope (Nikon, Japan) under fluorescent light using UV-2A filters.

### 3.2. Immunohistochemical Examination

For immunohistochemical examinations, healthy and carious molars were split using mechanical methods; the pulp was then extracted from the tooth chamber and fixed in 4% buffered formalin (pH = 7.4) for 12 h. The material was then decalcified in an alcohol series and embedded in paraffin. Four µm thick strips were subjected to the immunohistochemical procedure. Paraffin preparations were used to obtain strips of 4 µm thickness to make histological preparations. The immunohistochemical procedure was applied to the strips. After rehydration, endogenous peroxidase (Peroxidase Blocking Reagent) (DAKO, Konin, Poland) was blocked. The strips were then washed twice for 5 min in distilled water and then digested with proteinase K (DAKO, Konin, Poland) and washed twice for 5 min in distilled water. Antigens were unmasked using an antigen unmasking solution (Vector, Burlingame, CA, USA), followed by the application of specific primary antibodies (Table 2) at a dilution of 1/500. After applying secondary antibodies Anti-Prox 1, Anti-VEGF Receptor-3, Anti-LYVE1, Anti-Collagen IV, and Anti-PDPN, staining with DAB+ substrate buffer and DAB+ chromogen visualisation system (DAKO, Konin, Poland) was performed, and haematoxylin (Mayer, Heidenheim, Germany) was used to stain cell nuclei. The antibodies listed in Table 2 were utilised for immunohistochemical examinations. A blank test, i.e., Anti-Collagen IV, was performed for each antibody. Staining intensity was evaluated following a 0–2 scale: 0—no reaction, 1—weak reaction, 2—strong reaction.

### 3.3. Scanning Electron Microscope Examination

After cutting the extracted teeth in the horizontal plane with a diamond saw, they were fixed in 2.5% glutaraldehyde with phosphate buffer, pH = 7.4. Samples were washed with phosphate buffer and dehydrated in increasing acetone concentrations (from 50% to 100%). The teeth were dried at the critical point, glued to the tables using conductive tape and sputtered with gold. Surface topography was examined at a 3000× and 5000× magnification.

## 4. Results

### 4.1. Light Microscope Examination

The histological examination assessed vascular development and the degree of dental pulp inflammation. It also identified potential angiogenesis or lymphangiogenesis sites and the nature of the infiltration. The light microscope evaluation of tooth preparations is given in Figure 1, Figure 2, Figure 3 and Figure 4.

Teeth with no signs of inflammation or other pathological lesions were marked as healthy and assigned to Group I (Figure 1). In some (Group II) teeth with neutrophilic infiltrate, no angiogenesis was observed (Figure 2). In contrast, angiogenesis with possible lymphangiogenesis was observed in (Group III) teeth with mixed lymphocytic and neutrophilic infiltration and macrophages (Figure 3). In other teeth, wide and numerous blood-filled vessels within the dental pulp were found but without a significant number of inflammatory cells (Group IV, Figure 4).

### 4.2. Fluorescence Microscope Examination

Evaluation of tooth preparations under fluorescence microscope is given in Figure 5, Figure 6, Figure 7 and Figure 8. In healthy teeth (Group I), besides standard erythrocyte-filled capillaries, some spaces were not filled with blood, with a poorly visible wall lined only with cells of single-layer squamous epithelium. These could be sinusoidal or potential capillaries. The element that differentiates them from regular vessels is that other cells are not present in their lumen. Taking into account anastomoses and the possibility of regulating blood flow through the dental pulp, these may be temporary vessels, so to speak, cut off from regular blood circulation (Figure 5). Various types of vessels were observed in Group II (Figure 6).

Groups III and IV (Figure 7 and Figure 8) showed numerous areas of very active angiogenesis, and Group IV (Figure 8) was free of inflammatory infiltration between newly formed vessels. The morphological examination could not determine the type of blood vessel. Instead, it evaluated the inflammation intensity in the pulp, with the strongest found in Groups III and IV and much weaker in Group II.

### 4.3. Immunohistochemical Examination

A light microscope was used to analyse the immunohistochemical reaction intensity. Microscopic images are shown in Figure 9, Figure 10, Figure 11, Figure 12, Figure 13, Figure 14, Figure 15, Figure 16, Figure 17, Figure 18 and Figure 19 and Table 3. The examination did not provide an unambiguous answer on the relationship between lymphangiogenesis and the degree of inflammation. However, the observed positive antibody reaction may indicate that lymphatic vessels are present.

Assessment of the immunohistochemical reaction intensity is presented in Table 3. No reaction stronger than grade 2 was observed during the examination.

### 4.4. Scanning Electron Microscope Examination

A scanning electron microscope was used to examine tooth surfaces within the pulp and dentine. Surface images from individual study groups are presented in Figure 19, Figure 20, Figure 21 and Figure 22.

Mechanical cracks in the dentine on the side of the pulp were observed on the surface of the healthy tooth, and extensive blood vessels were present in the pulp (Group I, Figure 19).

Partial dentine demineralization was observed in Group II (Figure 21). The odontoblast layer removal and partial exposure of dentinal tubules revealed largely damaged and undirected collagen fibres. Blood vessels were faintly visible.

Vessels with thickened walls were frequently observed in Group III. These were the sites to which the newly formed blood vessels of the pulp were attaching. Numerous macrophages, lymphocytes and neutrophils were visible among these vessels.

Wide and numerous blood vessels were observed in Group IV (Figure 22).

## 5. Discussion

Knowledge of the lymphatic system structure makes it possible to determine the ways diseases spread, especially inflammatory ones. The normal passage of substances into tissues depends on the interrelated circulation of blood and lymph. Lymph plays an important role in preventing the spread of external factors, carrying its contents through the lymph nodes, which are a specific filter that traps foreign particles and microorganisms. Specific immunohistochemical methods make it possible to credibly determine the presence of lymphatic endothelium in the dental pulp. Commercially available immunohistochemical markers are specific to lymphatic vessel endothelium. Using them in combination with an electron microscope and enzyme-histochemical procedures makes it possible to confirm the presence of lymphatic vessels.

This paper aims to demonstrate lymphatic vessels in the dental pulp under physiological conditions and in teeth with caries-induced inflammation using microscopic and immunohistochemical methods.

Our study of Group I, which consisted of healthy teeth, with a light microscope, revealed the presence of spaces without erythrocytes. They were only lined with simple squamous epithelium cells. These could be sinusoidal or potential capillary vessels. However, due to anastomoses and the possibility of regulating blood flow through the pulp, they may be vessels temporarily detached from regular blood flow. Numerous areas of very active angiogenesis were observed in carious teeth. In Group IV, no inflammatory infiltrations were observed between newly formed vessels in teeth previously treated for caries. The morphological examination assessed the inflammation intensity, with the strongest in Groups III and IV and much weaker in Group II. Groups II, III and IV included carious teeth. In Group II, neutrophil infiltration and lymphocytic and neutrophilic infiltration with macrophages were present. Group IV consisted of teeth previously treated for caries, with wide and numerous blood-filled vessels within the dental pulp and without a significant number of inflammatory cells. In Groups II–IV, it was impossible to determine lymphatic vessel presence based on morphological features. The presence of lymphatic vessels in healthy teeth was confirmed under a light and fluorescence microscope.

The examination of the tooth surface topography under a scanning electron microscope revealed changes within the pulp and dentine in individual groups. Mechanical fractures within the dentine on the side of the pulp and extensive blood vessels in the pulp were observed in Group I. Dentine demineralization was observed in Groups II–IV, and blood vessels were also visible. In addition, numerous macrophages, lymphocytes and neutrophils were detected in Group II.

The immunohistochemical examination (IHC) using specific endothelial lymphatic vessel antibodies (Anti-Prox 1, Anti-VEGFR-3, Anti-LYVE1, Anti-PDPN) and the control sample testing using Anti-Collagen IV antibodies in material from Groups I–IV did not provide an unambiguous answer on the relationship between lymphangiogenesis indicators and the type or degree of dental inflammation. The fixation time is short during the immunohistochemical examination, and this process should be observed. In our study, decalcification was omitted because it is associated with denaturation or damage to protein structure. Decalcification that lasts for too long leads to a degradation of collagen and thus a degradation of both tooth wall and pulp, including blood vessels, which impacts the IHC results. Such proteins will no longer be antibody specific, which may lead to false results.

The difference in results may stem from a different material preparation for testing. This study used a diamond saw to immediately expose the pulp, harvest it and prepare it for staining, which enabled IHC analysis of the vascular system without lengthy procedures.

In their study using enzyme histochemistry, Aoyama et al. [9] observed that larger lymphatic vessels are located in the central part of the pulp, while smaller ones are in the peripheral regions. The authors believe that lymphatic drainage begins beneath the odontoblast layer from blind-ended lymphatic vessels that run apically and merge into larger collective vessels. They run parallel to blood vessels and capillaries. They leave the pulp through the apical foramen, while minor lymphatic vessels may exit through lateral canals and additional foramina.

Berggreen et al. [11] share the views of Aoyama et al. [9] regarding the distribution of lymphatic vessels in the dental pulp. They believe that the vast majority of these vessels are located in the central part of the pulp, while a small number are under the odontoblast layer. They also noticed that lymphatic vessels could exit the dental pulp in the furcation area through the lateral canals within the root. In their IHC study, Bergreen et al. observed horizontally running lymphatic vessels in the interdental septa that connect the periodontium of adjacent teeth. They concluded that these connections might enable the spread of infections from periodontal ligaments. Typical vessel locations were not observed in the analysis of incisors, molars and premolars. However, larger vessels were located in the root region, while more numerous capillaries were observed in the crown region. It should be emphasised that both types of vessels, large and small, were located in both areas.

In this study, light and fluorescence microscope examinations confirmed the presence of vessels with morphological features of lymphatic vessels. Light and electron microscope studies by Bishop and Malhotra [7] revealed differences in the distribution of lymphatic vessels in individual groups of feline teeth. They found no lymphatic vessels in the odontoblast layer and pulp horns. Berggreen et al. [11] confirmed the absence of lymphatic vessels in the odontoblast layer but detected lymphatic vessels in the pulp horns. Bishop and Malhotra [7] speculated that lymphatic vessels exit the pulp through lateral root canals, as an increasing number of lymphatic vessels were discovered in cross-sections of teeth stretching from the central part to the apex of the dental root.

The different arrangements of connective tissue in the dental pulp may promote the presence of lymphatic vessels or the formation of extravascular lymphatic pathways in the form of tissue fissures. Oehmke et al. [12] used a light and electron microscope to trace the flow of patent blue in dental preparations. They suggested that empty spaces surrounded by cells similar to vascular endothelial cells were present in the dental pulp. Our analysis revealed the presence of large, extensive spaces resembling sinusoidal vessels, not in communication with blood vessels. Their presence was observed in the entire dental cavity, not only at the edges. These studies indicate extravascular means of clearing fluids and macromolecules. It was observed that the lymph found in the coronal portion is taken up by the intercellular fluid and filtered by lymphatic vessels in the dental root apex region. It cannot be straightforwardly determined if these fissures are part of the venous or lymphatic system.

The healthy teeth analysis did not reveal angiogenesis or a specific lymphatic system in the periapical region of the dental root. The root periapical area revealed a large number of lymphatic vessels bound to periodontal ligaments, which could connect with the lymphatic vessels of the dental pulp [11,13,14,15]. Heyeraas et al. [16] confirmed the presence of extravascular fluid and protein drainage pathways using I-labelled albumin in feline teeth. Tracking of the distribution of the labelled albumins clearly indicated the presence of extravascular transport of fluids.

Our study revealed that Anti-Prox1 was expressed in healthy teeth but not in carious teeth; however, it should be noted that equine teeth are much larger and have a different structure. Staszyk et al. [4] investigated the presence of lymphatic vessels in the equine periodontium: the gingivae and the trabecular bone of the mandible and maxilla. They observed blood vessels without lymphatic vessels in the cementum of the tooth. Samples were processed for immunoreaction with Anti-Prox1.

In this study, both receptors were present in healthy teeth and teeth with caries alike. Berggreen et al. hypothesised that the lymphatic system is present in dental tissues, while lymphangiogenesis occurs to enable transport of fluids, proteins and external factors during inflammation, as well as to provide transport to the antigen-presenting cells in regional lymphatic vessels. An immunohistochemical examination using the LYVE-1 and VEGFR-3 markers revealed a different system of lymphatic vessels in rat incisors and molars. Lymphatic vessels in the periodontal ligament were aligned horizontally, together with the periodontal ligaments of the adjacent tooth, constituting a potential route of infection through the bone between adjacent teeth [11]. This observation indicates that periodontal or pulp infections can spread from one tooth to another through the interdental bone via lymphatic vessels.

Lohberg et al. demonstrated that murine dental pulp does not contain initial lymphatic vessels [17]. Using lymphocyte-specific LYVE-1 and podoplanin (D2-40) antibodies, the authors examined decalcified paraffin-embedded mouse heads and stained serial cross-sectional products. They did not find lymphatic vessels in the dental pulp, but they did observe branched cells, weakly positive for LYVE-1, which could represent dendritic cells or macrophages. However, the interstitial fluid is naturally drained into the lymphatic vessels accompanying dental arteries and nerves. However, the results may have been distorted because the material was kept in decalcifying fluids for a long time. It should be noted that murine teeth are relatively small, and the interstitial fluid produced by them does not have to pass through the lymphatic system.

Gerli et al. [18] advanced the thesis that lymphatic vessels are not present in the pulp of healthy teeth but may appear as a result of inflammation. Under normal conditions, interstitial fluid that was not reabsorbed by veins would flow through endothelial channels [1] from the coronal region of the tooth through the extracellular matrix into the root canal and leave the tooth through the apical foramina. Anti-LYVE1 was not expressed in pulp with inflammation caused by carious lesions, although other lymphatic endothelial markers were expressed. Perhaps some vessels are transformed or fused with blood vessels due to severe carious lesions.

LYVE-1 positive (LYVE-1+) and VEGFR-3 positive (VEGFR-3+) immune cells can be found in the pulp and derive from the monocyte–macrophage (M/M) lineage. However, these cells were absent in inflamed pulp with concurrent tooth decay. Macrophages may contribute to lymphangiogenesis by directly forming lymphatic vessels due to inflammatory lesions in the pulp [11]. In vitro, macrophages treated with anti-inflammatory agents form tube-like structures that mimic lymphatic vessels. This seems unlikely in the case of dental pulp.

It was also observed that novel LYVE-1+ and VEGFR-3+ populations were present in both normal and infected pulp. The LYVE-1+ cells in the pulp originated from the monocyte–macrophage lineage. Interestingly, LYVE-1+ macrophages in healthy tissues were described in murine sclera, choroid and iris [19]. The role of these immune cells in the eye is unknown, but it was demonstrated that a subpopulation of these cells, Sca+ (stem cell antigen-positive), confirms that they are progenitor cells for haematopoietic lymphatic cells [19]. It was reported that CD45+ and CD34+ antigens were present in the LYVE-1+ cells in the pulp. Haematopoietic progenitor cells are expressed by both of these markers, which implies that the LYVE-1 cells may be their source [4,20].

A dense network of blindly terminated lymphatic vessels enables fluid flow from the coronal portion of rat molars, while in incisors, the fluid is transported to the apical region of the lymphatic vessels due to pressure difference. These differences may stem from the fact that rat incisors constantly grow, unlike rat molars [11].

Sawa et al. [21] revealed that lymphatic vessels were located in the central part of the pulp and its periphery. It was confirmed by Aoyama et al. [9] and our research. It indicates that lymphatic drainage of the pulp of the human tooth starts from its periphery and accumulates in its central part. Inflammation of the pulp also begins in its periphery. Sawa et al. [21] used monoclonal antibodies specific for the human thoracic duct and an antiserum against human laminin to identify lymphatic vessels in the dental pulp. However, these antibodies were not specific.

In our study involving the immunohistochemical method, we used specific lymphatic vascular endothelial antibodies (Anti-Prox 1, Anti-VEGFR-3, Anti-LYVE1, Anti-PDPN) and Anti-Collagen IV antibodies as control. To date, no such combination of antibodies has been used simultaneously. However, this study did not confirm the association between lymphangiogenesis indicators and the type or degree of dental inflammation.

Perhaps more detailed studies using molecular markers to identify lymphatic vessels in the tooth will make it possible to verify their presence and function in the future. Lymphatic endothelial heterogeneity needs to be addressed in further studies concerning determining the phenotypic appearance and gene expression profile.

## 6. Conclusions

Based on our studies and results, it can be concluded that there was a moderate correlation between pulp inflammation and the formation of new vessels, including lymphatic vessels. In view of the above, it can be said that as inflammation increases, the size of the vascular bed that enables circulation of body fluids, blood and lymph increases as well. However, microscopic and immunohistochemical studies did not conclusively demonstrate if these vessels form systems within the pulp that facilitate drainage of the tooth cavity.

## Figures and Tables

**Figure 1 biology-11-00635-f001:**
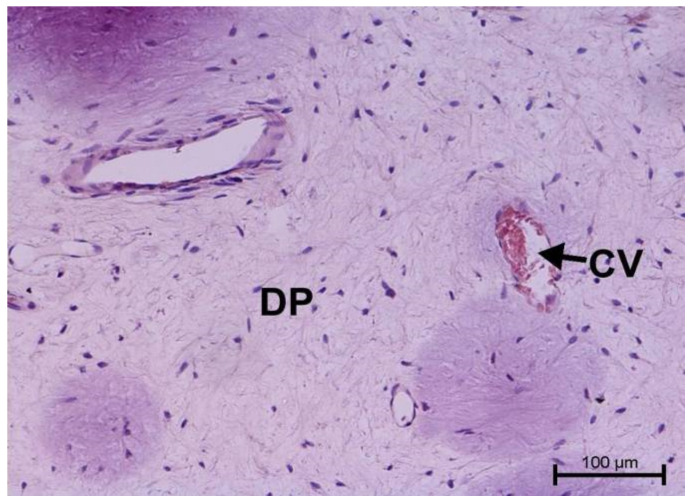
Light microscope image, human molar—Group I. Dental pulp with blood vessels. DP—dental pulp, CV—capillary vessel. HE stain. Mag. 200×.

**Figure 2 biology-11-00635-f002:**
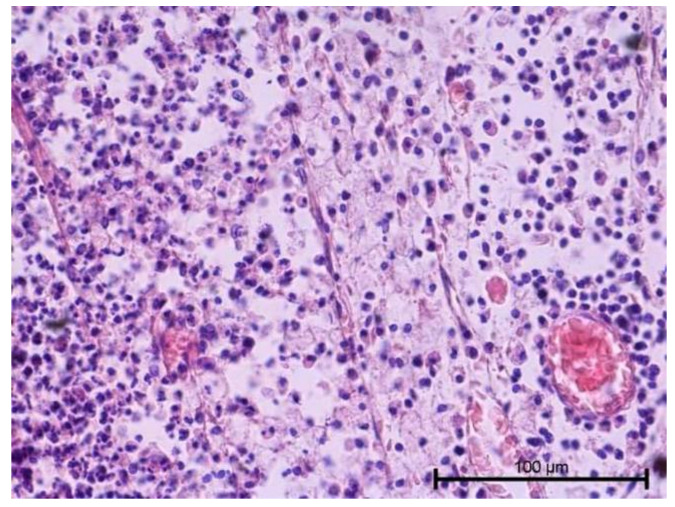
Light microscope image, human molar after extraction—Group II. Mild pulpitis. Visible neutrophilic infiltration and minor collapsed capillaries. HE stain. Mag. 200×.

**Figure 3 biology-11-00635-f003:**
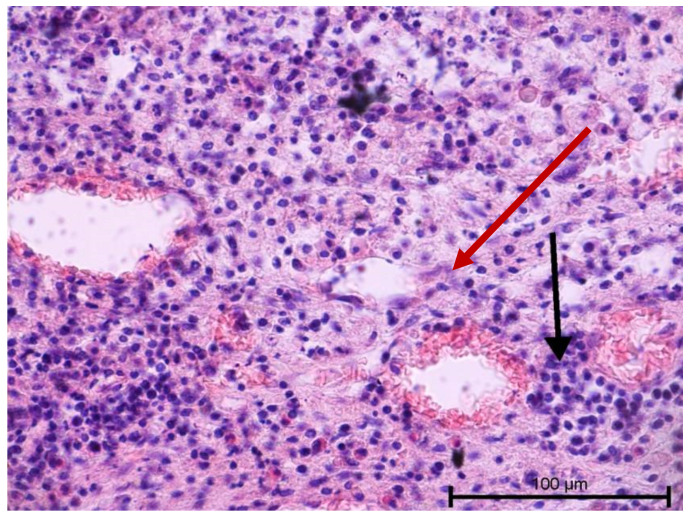
Light microscope image, human molar after extraction—Group III. Lymphocytice infiltrations (black arrow) and small capillary vessel (red arrow). Stain HE. Mag. 200×.

**Figure 4 biology-11-00635-f004:**
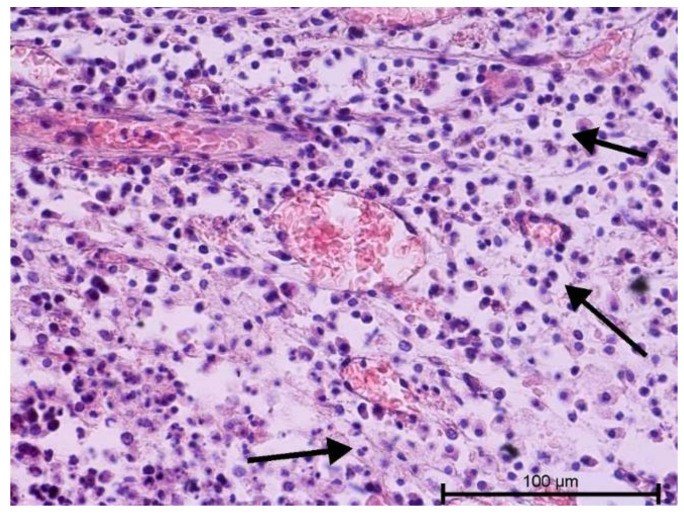
Light microscope image, human molar after extraction—Group IV. Very intense angiogenesis (arrow). HE stain. Mag. 200×.

**Figure 5 biology-11-00635-f005:**
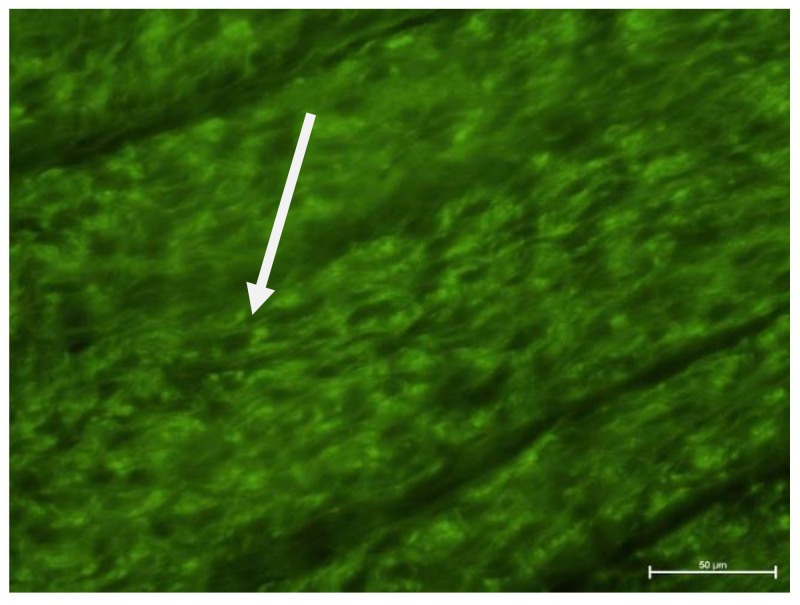
Fluorescence microscope image, human molar tooth—Group I. Visible capillaries that do not contain erythrocytes. UV-2A. Mag. 200×.

**Figure 6 biology-11-00635-f006:**
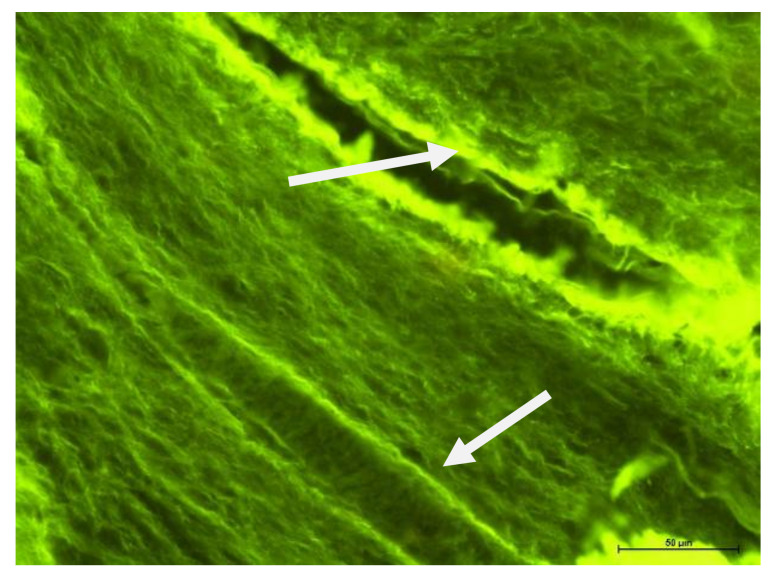
Fluorescence microscope image, human molar—Group II. Visible different types of vessels (arrow) in different sections. UV-2A. Mag. 200×.

**Figure 7 biology-11-00635-f007:**
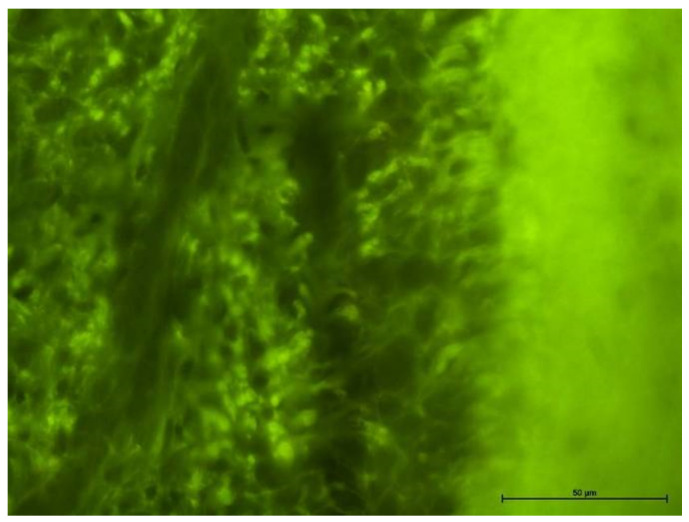
Fluorescence microscope image, human molar—Group III. Visible angiogenesis (arrow) in the dentin region (Z). UV-2A. Mag. 200×.

**Figure 8 biology-11-00635-f008:**
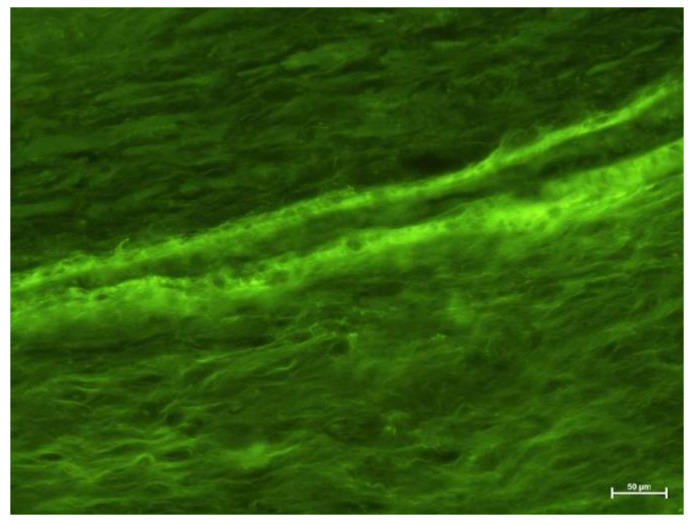
Fluorescence microscope image, human molar—Group IV. Typical angiogenesis with a new blood vessel formation in the dental pulp. UV-2A. Mag. 200×.

**Figure 9 biology-11-00635-f009:**
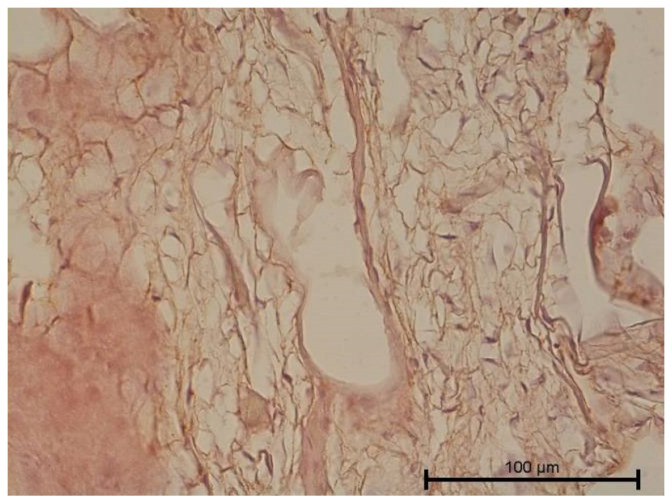
Light microscope image. Healthy tooth. Anti-PDPN—positive. Single infiltrating cells. Mag. 400×.

**Figure 10 biology-11-00635-f010:**
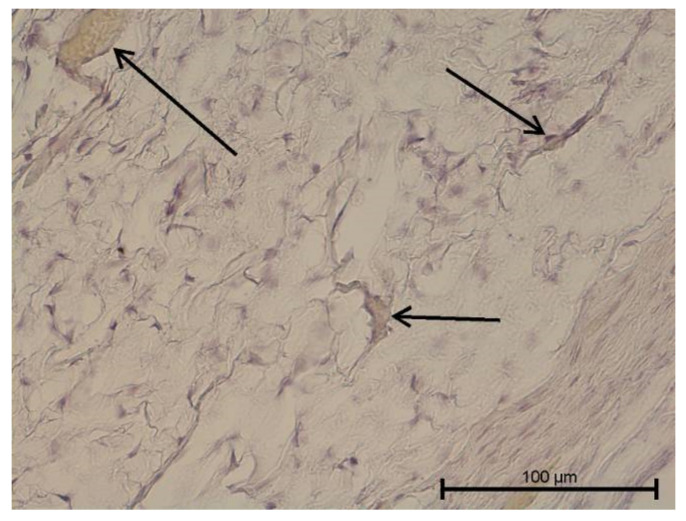
Light microscope image. Carious tooth. Anti-PDPN—positive (black arrow). Single leukocytes visible. Mag. 400×.

**Figure 11 biology-11-00635-f011:**
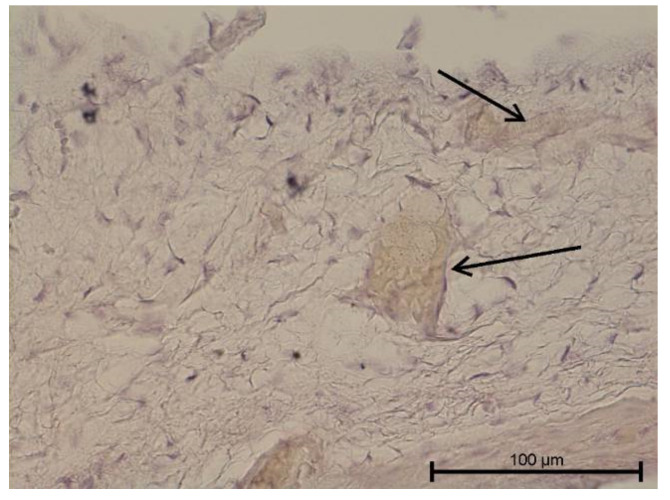
Light microscope image. Healthy tooth. Anti-Collagen IV—positive (black arrow). Mag. 400×.

**Figure 12 biology-11-00635-f012:**
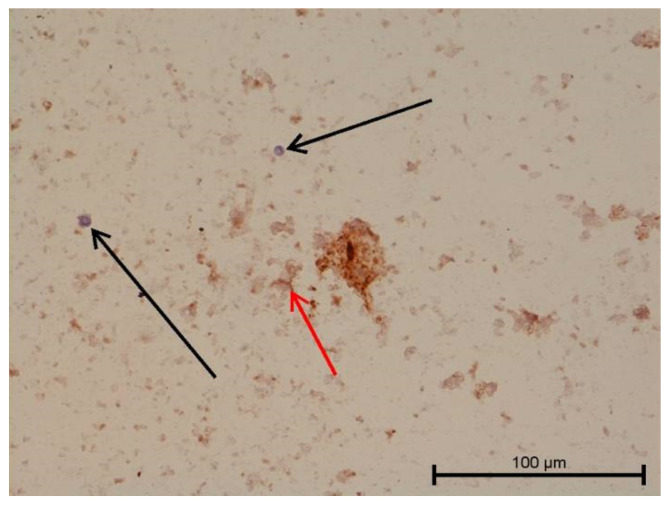
Light microscope image. Carious tooth. Anti-Collagen IV—positive (red arrow). Severe inflammation neutrophils present (black arrow). Mag. 400×.

**Figure 13 biology-11-00635-f013:**
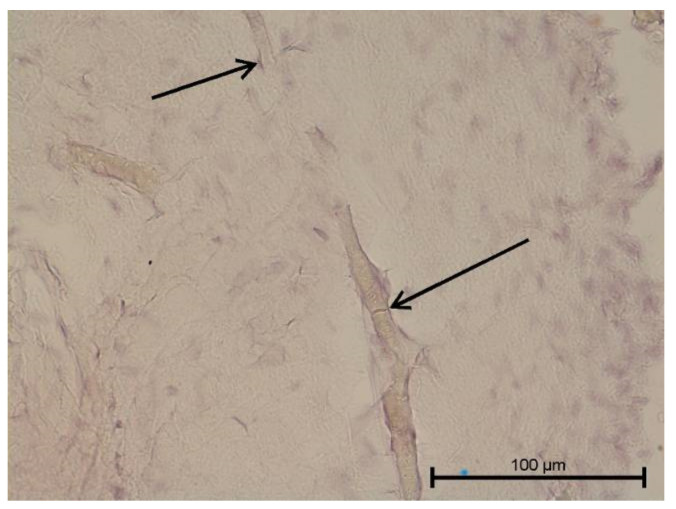
Light microscope image. Healthy tooth. Collagen IV—positive (black arrow). Mag. 400×.

**Figure 14 biology-11-00635-f014:**
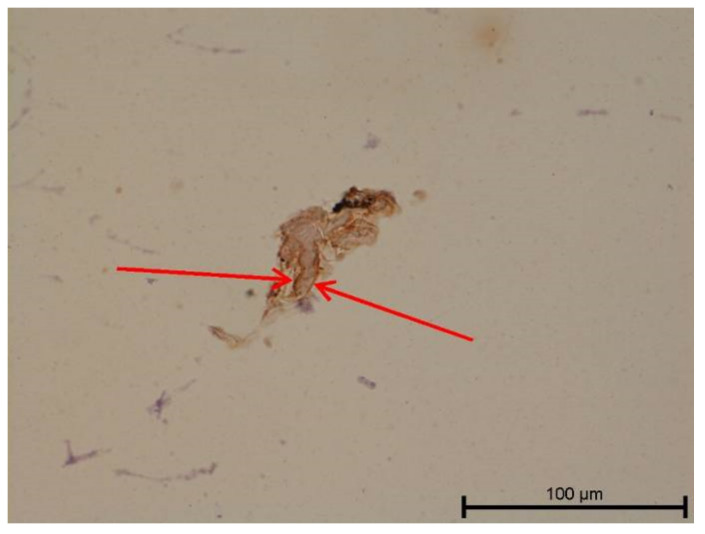
Light microscope image. Carious tooth. Anti-PDPN—positive. Inflammatory infiltration (red arrows). Mag. 400×.

**Figure 15 biology-11-00635-f015:**
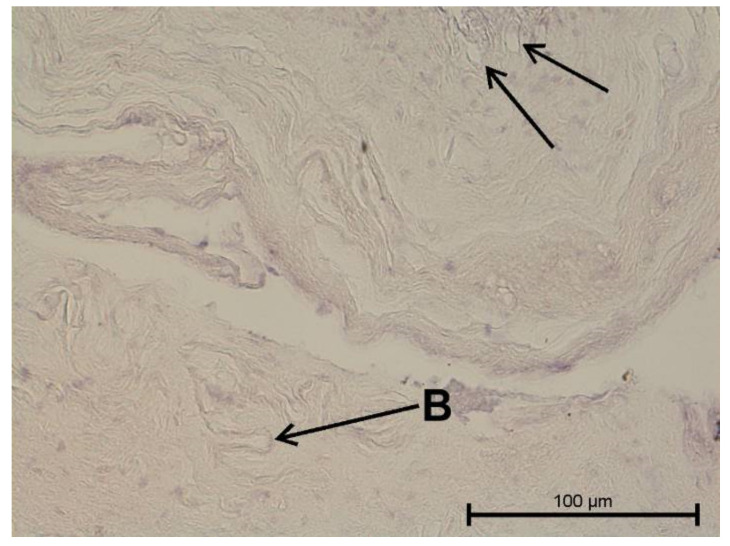
Light microscope image. Healthy tooth. Anti-Prox 1—negative (black arrow). B-visible newly formed blood vessels as a result of angiogenesis. Mag. 400×.

**Figure 16 biology-11-00635-f016:**
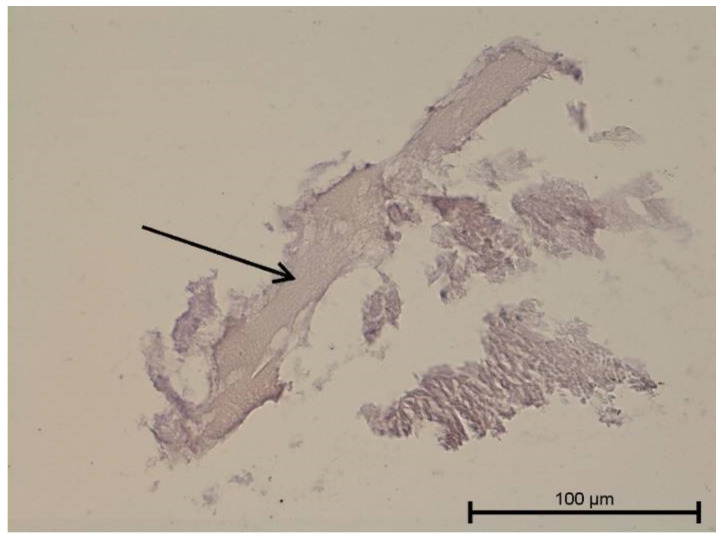
Light microscope image. Carious tooth. Anti-VEGFR 3—weak positive (black arrow). Mag. 400×.

**Figure 17 biology-11-00635-f017:**
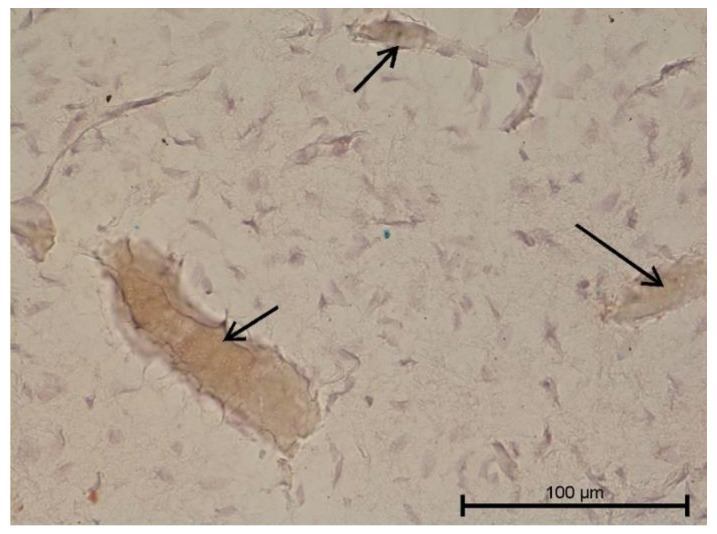
Light microscope image. Healthy tooth. Anti-VEGFR 3—positive (black arrow). Tooth and blood vessel cells. Mag. 400×.

**Figure 18 biology-11-00635-f018:**
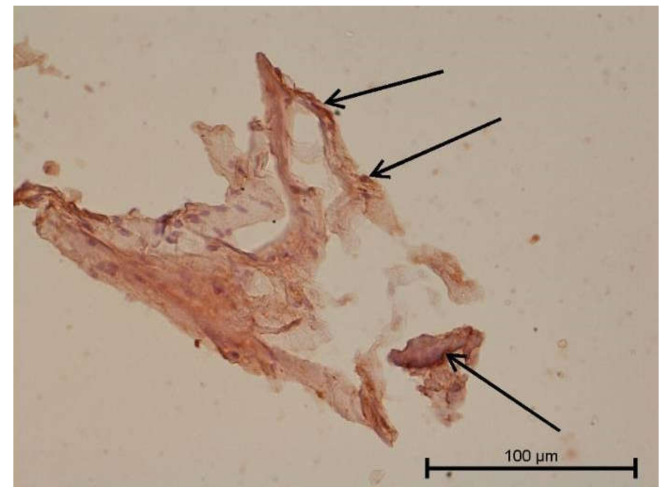
Light microscope image. Healthy tooth. Anti-VEGFR 3—positive (black arrow). Mag. 400×.

**Figure 19 biology-11-00635-f019:**
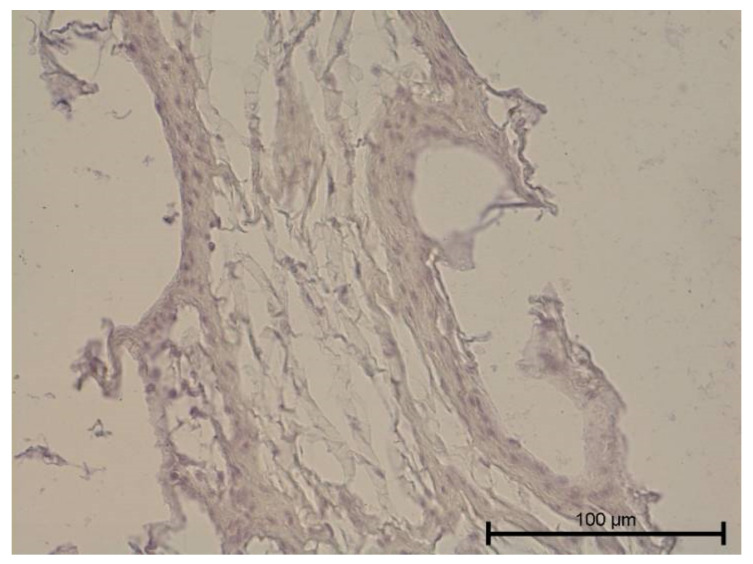
Light microscope image. Carious tooth. LYVE1—negative. Mag. 400×.

**Figure 20 biology-11-00635-f020:**
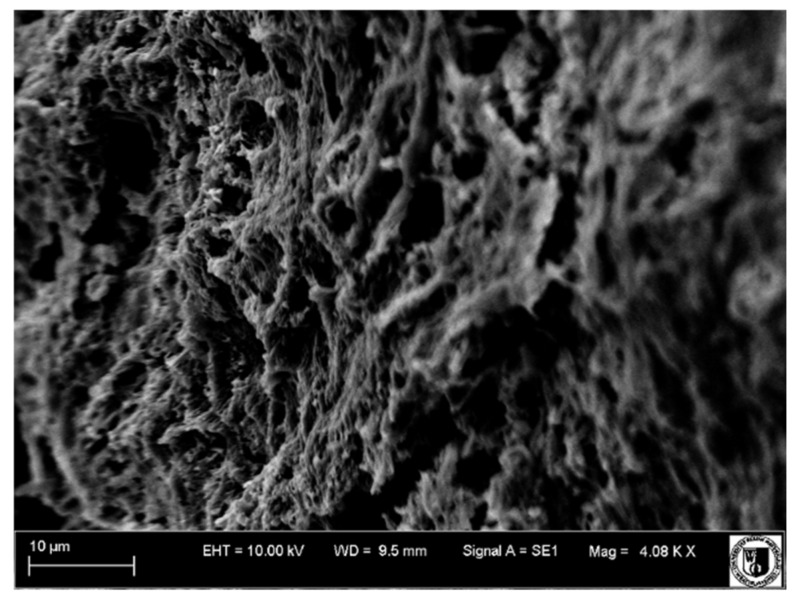
Scanning electron microscope image, human molar—Group II. Slightly damaged collagen fibres of the pulp run disorderly in different directions. 4000×.

**Figure 21 biology-11-00635-f021:**
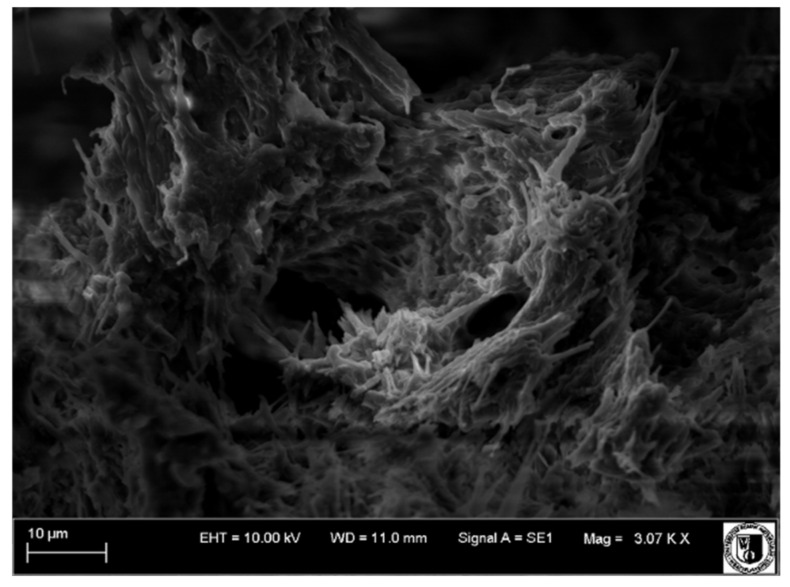
Scanning electron microscope image, human molar—Group III. Delaminated, oedematous blood vessels (arrow). 3000×.

**Figure 22 biology-11-00635-f022:**
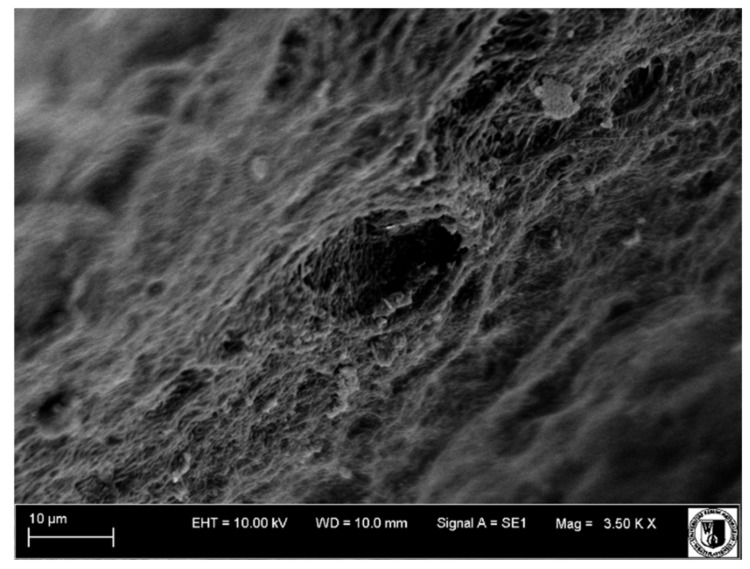
Scanning electron microscope image, human molar—Group IV. Minor vessel in the dental pulp (arrow). 3500×.

**Table 1 biology-11-00635-t001:** The number of teeth used for examinations.

Examinations	Healthy Teeth [*n*]	Carious Teeth [*n*]
Histological under light microscope	5	10
Immunohistochemical	5	10
Histological under a fluorescence microscope	5	10
Scanning electron microscope	5	10

**Table 2 biology-11-00635-t002:** Antibodies used in the IHC test.

Antibody	Manufacturer	Type of Antibody
Anti-Prox 1	Sigma-Aldrich, Germany	human, polyclonal
Anti-VEGFR-3	Sigma-Aldrich, Germany	human, monoclonal
Anti-LYVE1	Sigma-Aldrich, Germany	human, polyclonal
Anti-Collagen IV	Sigma-Aldrich, Germany	human, polyclonal
Anti-PDPN	Sigma-Aldrich, Germany	human, polyclonal

**Table 3 biology-11-00635-t003:** Staining scale scores for specific antibodies.

Antibody	Healthy Teeth X ± SD	Teeth with Caries X ± SD
Anti-PDPN	0.75 ± 0.19	1.25 ± 0.19
Anti-Collagen IV	1.75 ± 0.19	1.25 ± 0.19
Anti-Prox 1	1.25 ± 0.22	0.00
Anti-VEGF Receptor-3	0.00	0.50 ± 0.22
Anti-LYVE1	0.00	0.00

Staining scale: 0–2, where 0—no reaction, 1—weak reaction, 2—strong reaction.

## Data Availability

No new data were created or analysed in this study. Data sharing is not applicable to this article.

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
