# Peer review of "Detection of Lymphatic Vessels in Dental Pulp"

_biology, 2022, doi:10.3390/biology11050635_

Round 1
Reviewer 1 Report
Is there a correlation between the evolution of inflammation and the formation of the lymphatic network?
Can the degree of root formation influence this process?
Author Response
Is there a correlation between the evolution of inflammation and the formation of the lymphatic network?
No clear correlation was detected. The observation of lymphatic vessels formation as result of inflammation was disputable.
Can the degree of root formation influence this process?
Yes, but in analysed teeth, root formation process was already halved
Reviewer 2 Report
Lymphangiogenesis occurs in an inflamed pulp. If lymphangiogenesis is defined as the development of lymphatic vessels from already existing ones, such a mechanism is possible only when lymphatic vessels are present in healthy teeth. The aim of the paper was to identify lymphatic vessels in the dental pulp using microscopic, immunohistochemical techniques under physiological and pathological conditions. The tissue material consisted of human teeth that were to be extracted. The authors observed that there was a moderate correlation between pulp inflammation and the formation of new vessels, including lymphatic vessels.
Overall the paper is very interesting, the conclusion are supported by the data obtained.
major concerns:
abstract section:
it is strongly suggested to modify the abstract and to rewrite the abstract better in english, the second sentence with " If....please revised.
Figure 3. Light microscope image, human molar after extraction ‐ group III. Lymphocytic‐neutrophilic infiltrations (black arrow) angiogenesis (red arrow).. Taking into consideration the paper entitled "NF-κB involvement in hyperoxia-induced myocardial damage in newborn rat hearts" and "In the carotid body, galanin is a signal for neurogenesis in young, and for neurodegeneration in the old and in drug-addicted subjects" how to perform the IHC analysis and how to insert the inset, please revised your figures adding inset to evindence in particular in this picture the neutrophilic infiltrations.
Figure 13. Microscopic image under a light microscope. Healthy tooth. Collagen IV ‐ positive result....ANTI-Collahen IV, please revise
(black arrow). Mag. 400×.
In all the figures done by IHC add a graph which rappresents the densitometric analysis.
The paper need to be revised extensively by a native english speaker.
Author Response
Lymphangiogenesis occurs in an inflamed pulp. If lymphangiogenesis is defined as the development of lymphatic vessels from already existing ones, such a mechanism is possible only when lymphatic vessels are present in healthy teeth. The aim of the paper was to identify lymphatic vessels in the dental pulp using microscopic, immunohistochemical techniques under physiological and pathological conditions. The tissue material consisted of human teeth that were to be extracted. The authors observed that there was a moderate correlation between pulp inflammation and the formation of new vessels, including lymphatic vessels.
Overall the paper is very interesting, the conclusion are supported by the data obtained.
major concerns:
abstract section:
it is strongly suggested to modify the abstract and to rewrite the abstract better in english, the second sentence with " If....please revised.
The text was revised.
Figure 3. Light microscope image, human molar after extraction ‐ group III. Lymphocytic‐neutrophilic infiltrations (black arrow) angiogenesis (red arrow).. Taking into consideration the paper entitled "NF-κB involvement in hyperoxia-induced myocardial damage in newborn rat hearts" and "In the carotid body, galanin is a signal for neurogenesis in young, and for neurodegeneration in the old and in drug-addicted subjects" how to perform the IHC analysis and how to insert the inset, please revised your figures adding inset to evindence in particular in this picture the neutrophilic infiltrations.
The description was modified to the lymphocytic infiltration only.
Figure 13. Microscopic image under a light microscope. Healthy tooth. Collagen IV ‐ positive result....ANTI-Collahen IV, please revise
(black arrow). Mag. 400×.
The text was revised.
In all the figures done by IHC add a graph which rappresents the densitometric analysis.
The paper need to be revised extensively by a native english speaker.
The text was revised.
Reviewer 3 Report
Thank for giving me an opportunity to review your interesting work.
The role of lymphatics in the dental pulp may be underestimated. So, the investigation of the lymphatics in the dental pulp is important.
But, I have some questions and concerns in the current manuscript as follows:
- In figure 3, you indicate angiogenesis by using red arrow. Why can you regard this blood vessel as angiogenesis? The immunostaining with the markers of angiogenesis is necessary.
- The quality of fluorescence microscopic images (fig 5-8) and immunohistochemistry (fig 9-19) are so poor. You describe in the result section as follows: In groups III and IV (Fig. 7-8) numerous areas of very active angiogenesis were observed, with group IV (Fig. 8) free of inflammatory infiltrations between newly formed vessels. You should count the number of blood or lymphatic lumens in each sample.
- In the evaluation of signal intensity in the immunohistochemical examination, how many doctors evaluated? Was evaluation blind?
- In figure 9 (I think that this is a representative photo), the signal intensity of PDPN is strong in the healthy tooth. But, the intensity of healthy tooth is weaker than one of tooth with caries. The signal intensity of PDPN is stronger in figure 9 than in figure 10.
- In the SEM images, the blood or lymphatic vessels and the cell-cell junctions of endothelial cells should be indicated with arrows.
Author Response
Thank for giving me an opportunity to review your interesting work.
The role of lymphatics in the dental pulp may be underestimated. So, the investigation of the lymphatics in the dental pulp is important.
But, I have some questions and concerns in the current manuscript as follows:
- In figure 3, you indicate angiogenesis by using red arrow. Why can you regard this blood vessel as angiogenesis? The immunostaining with the markers of angiogenesis is necessary.
The changes were made in description of the picture
- The quality of fluorescence microscopic images (fig 5-8) and immunohistochemistry (fig 9-19) are so poor. You describe in the result section as follows: In groups III and IV (Fig. 7-8) numerous areas of very active angiogenesis were observed, with group IV (Fig. 8) free of inflammatory infiltrations between newly formed vessels. You should count the number of blood or lymphatic lumens in each sample.
In situation of the inflammation, single capillary vessels were not observed. Calculation of the capillary bed in such conditions was impossible.
- In the evaluation of signal intensity in the immunohistochemical examination, how many doctors evaluated? Was evaluation blind?
Yes. The results were obtained basing upon 3 independent observations, which were compared.
- In figure 9 (I think that this is a representative photo), the signal intensity of PDPN is strong in the healthy tooth. But, the intensity of healthy tooth is weaker than one of tooth with caries. The signal intensity of PDPN is stronger in figure 9 than in figure 10.
The process of immunohistochemistry was performed at the same time, however pictures ware taken in the automatic mode, therefore may have different intensity.
- In the SEM images, the blood or lymphatic vessels and the cell-cell junctions of endothelial cells should be indicated with arrows.
Revised and fixed